# CREB1 Is Involved in miR-134-5p-Mediated Endometrial Stromal Cell Proliferation, Apoptosis, and Autophagy

**DOI:** 10.3390/cells12212554

**Published:** 2023-10-31

**Authors:** Xiaodan Li, Xiaolei Yao, Kang Li, Jiahe Guo, Kaiping Deng, Zhipeng Liu, Fan Yang, Yixuan Fan, Yingnan Yang, Huabin Zhu, Feng Wang

**Affiliations:** 1Hu Sheep Academy, Nanjing Agricultural University, Nanjing 210095, China; 2019205003@njau.edu.cn (X.L.);; 2Jiangsu Livestock Embryo Engineering Laboratory, Nanjing Agricultural University, Nanjing 210095, China; 3Embryo Biotechnology and Reproduction Laboratory, Institute of Animal Science, Chinese Academy of Agricultural Sciences, Beijing 100193, China

**Keywords:** endometrial receptivity, CREB1, miR-134-5p, endometrial stromal cells, proliferation, apoptosis, autophagy

## Abstract

The successful establishment of endometrial receptivity is a key factor in ensuring the fertility of ewes and their economic benefits. Hu sheep have attracted attention due to their high fecundity and year-round estrus. In this study, we found that in the luteal phase, the uterine gland density, uterine coefficient, and number of uterine caruncles of high-fertility Hu sheep were higher than those of low-fertility Hu sheep. Thousands of differentially expressed genes were identified in the endometrium of Hu sheep with different fertility potential using RNA sequencing (RNA-Seq). Several genes involved in endometrial receptivity were screened using bioinformatics analysis. The qRT-PCR analysis further revealed the differential expression of cAMP reactive element binding protein-1 (*CREB1*) in the Hu sheep endometrium during the estrous cycle. Functionally, our results suggested that *CREB1* significantly affected the expression level of endometrial receptivity marker genes, promoted cell proliferation by facilitating the transition from the G1 phase to the S phase, and inhibited cell apoptosis and autophagy. Moreover, we observed a negative linear correlation between miR-134-5p and *CREB1* in the endometrium. In addition, *CREB1* overexpression prevented the negative effect of miR-134-5p on endometrial stromal cell (ESC) growth. Taken together, these data indicated that *CREB1* was regulated by miR-134-5p and may promote the establishment of uterine receptivity by regulating the function of ESCs. Moreover, this study provides new theoretical references for identifying candidate genes associated with fertility.

## 1. Introduction

Fertility plays a pivotal role in the economic aspect of sheep production. Hu sheep, a native Chinese breed, have garnered attention for their precocious puberty, year-round estrus, and high fecundity (average litter size 2.06) [1,2]. To date, several major fecundity genes, including the bone morphogenetic protein receptor IB (*BMPR-IB*) gene [3,4], the bone morphogenetic protein 15 (*BMP15*) gene [5,6], and the growth differentiation factor 9 (*GDF9*) gene [7,8], have been discovered. Among these genes, *FecB* is recognized as the first major gene linked to prolificacy and multiple births in sheep. This arises from a point mutation at A746G in *BMPR-IB*, leading to an amino acid mutation from glutamine to arginine [9,10]. According to previously published literature, *FecB* mutations exert cumulation effects on ovulation number and litter size in sheep; for example, possessing one copy of *FecB* mutation increases ovulation number by 1.5 and litter size by 1, whereas having two copies of *FecB* mutation increases ovulation number by 3 and litter size by 1.5, respectively [11]. However, some Hu sheep carrying the *FecB^BB^* genotype also give birth to single lambs. Thus, we speculated that factors other than the BB genotype mutation of *BMPR-IB* may influence the litter size in Hu sheep.

Key processes in mammalian reproduction include ovarian follicular dynamics, ovulation, luteinization, luteolysis, and endometrial remodeling [12,13,14]. Research on ovine fecundity has largely focused on ovarian function [15,16,17]. However, uterine function also plays a critical role in prolificacy. The uterus is an important site for embryo implantation, and good endometrial receptivity and normal maternal fetal recognition are closely related to fertility. The sheep estrous cycle averages 16–17 days, comprising follicular and luteal phases distinguished by ovarian morphology and hormone secretion [18]. The endometrium undergoes a repeated cycle of proliferation, differentiation, and repair to prepare the embryo for successful implantation and maintenance of pregnancy under the control of the ovarian steroidal hormones estrogen (E2) and progesterone (P4) [19,20]. The endometrium is regenerated and differentiated in the follicular phase, whereas its metabolic demands shift in the luteal phase [21]. Throughout the preimplantation period, embryonic growth is sustained by maternal uterine secretions comprising a diverse array of nutrients including proteins, amino acids, pyruvate, lactate, carbohydrates, and fatty acids [22,23,24,25]. In addition, hormones, cytokines, transcription factors, adhesion molecules, and glycoproteins act on the endometrium, regulate endometrial function, and promote interactions between the endometrium and embryo [26,27,28,29,30,31,32]. The critical roles of the uterus in supporting embryo development highlight this organ as a potential source of valuable prolificacy genes. Therefore, a comprehensive understanding of the molecular mechanisms governing uterine-related functions is of paramount importance when investigating reproduction in female sheep.

Endometrial receptivity is a prerequisite for successful embryo implantation and pregnancy [33]. Studies have shown that unsound endometrial receptivity leads to embryo loss during preimplantation, which is an important reason for the low litter size of farm animals [34]. The transcription factor cAMP response element-binding protein-1 (*CREB1*), as an important target for multiple signaling pathways, was reported to be markedly dysregulated in endometriosis (EMS) patients [35,36]. Endometrial receptivity has been found to decline in patients clinically diagnosed with EMS [37]. Therefore, we speculate that dysregulation of *CREB1* may affect endometrial receptivity. In addition, endometrial thickness, uterine blood flow, and uterine gland development can be used as indicators of uterine receptivity [38,39]. Our previous studies compared the endocrinology and endometrial morphology of high-fertility Hu sheep (*FecB^BB^* genotype) and low-fertility Hu sheep (*FecB^BB^*/*FecB^B+^* genotype) in the follicular phase [40]. In the present study, we first analyzed the endocrinology and endometrial morphology of three groups of Hu sheep (LP_HBB, LP_LBB, and LP_LB+) in the luteal phase. Then, RNA sequencing (RNA-seq) was utilized to analyze the changes in gene expression profiles in Hu sheep endometrium during the luteal phase and follicle–luteum transformation process. *CREB1*, which has a targeting-binding relationship with miR-134-5p, significantly affected the expression level of endometrial receptivity marker genes, promoted the proliferation of endometrial stromal cells (ESCs), and inhibited their apoptosis and autophagy. Our results contribute to the overall improvement of sheep breeding in China and provide valuable insights into the regulatory mechanisms governing the fertility of Hu sheep. Moreover, elucidating the signals that control uterine receptivity and preimplantation could enable the diagnosis and identification of the causes underlying pregnancy loss. This knowledge can also be applied to enhance pregnancy rates in domestic animals and humans.

## 2. Results

### 2.1. Analysis of Hu Sheep Uterine Morphology and Endocrinology in Luteal Phase

Hu sheep in the LP_HBB group displayed greater uterine gland density compared to the LP_LBB and LP_LB+ groups (Figure 1A,C). The uterine coefficient was higher in the LP_HBB of the Hu sheep group than that in the LP_LB+ group but not in the LP_LBB group (Figure 1B). The number of uterine caruncles was higher in the LP_HBB of the Hu sheep group than that in the LP_LBB group but not in the LP_LB+ group (Figure 1I). There were no significant differences among the three groups, including endometrial microvascular D (MVD), endometrial ductal gland invaginations (DGI), luminal epithelium (LE) thickness, endometrial thickness, and myometrial thickness (Figure 1D–H). Moreover, the level of serum progesterone (P4) was lower in both the LP_HBB and LP_LBB groups of Hu sheep compared to those in the LP_LB+ group (Appendix A). In contrast to the trend for the level of P4, serum placental growth factor (PLGF) appeared to be highest in the LP_HBB group of sheep, followed by the LP_LBB group, and lowest in the LP_LB+ group (Appendix A). There were no differences in the serum levels of estrogen (E2) and prolactin (PRL) in the three groups of Hu sheep (Appendix A).

### 2.2. Characteristics of CREB1 during Estrous Cycle of Sheep

By sequencing, we found 378 (245 upregulated and 133 downregulated), 768 (421 upregulated and 347 downregulated), and 1413 (504 upregulated and 909 downregulated) DEGs in LP_HBB VS LP_LBB, LP_HBB VS LP_LB+, and LP_LBB VS LP_LB+, respectively (Appendix A). Functional annotation and analysis were performed using GO (Appendix A) and KEGG (Appendix A). In addition, we conducted a conjoint analysis of laboratory data from the follicular phase with novel data from the luteal phase and found that 9359 (2241 upregulated and 7118 downregulated), 8661 (1981 upregulated and 6680 downregulated), and 8743 (2185 upregulated and 6558 downregulated) DEGs in FP_HBB VS LP_HBB, FP_LBB VS LP_LBB, and FP_LB+ VS LP_LB+ groups, respectively (Appendix A), and functional annotation and analysis were performed using GO (Appendix A) and KEGG (Appendix A). These results reveal multiple biological processes and pathways that have the potential to contribute to endometrial receptivity establishment, including, in particular, the GO pathway, i.e., proteolysis, apoptosis, cell proliferation, and cell differentiation, as well as the KEGG pathway, i.e., cell cycle, MAPK, estrogen, and cGMP-PKG signaling pathway.

Some key node genes closely related to endometrial receptivity were screened by performing Protein–Protein Interaction Networks (PPI) analysis, including *BMP2*, *CREB1*, *ESR2*, *IGF2*, *PPP2R2A*, *PTGS2*, *SMAD1*, *SMAD2*, *STAT3*, and *WNT6* (Appendix A). Then, we detected their expression changes in the follicle-luteal transformation process using qRT-PCR and found that *BMP2*, *CREB1*, *ESR2*, *IGF2*, *PTGS2*, and *WNT6* were differentially expressed (Figure 2A). Among them, *CREB1* has been reported to be closely related to endometriosis and ovarian hormone secretion, and therefore, we selected *CREB1* as the object of the following research. The results of qRT-PCR also displayed that *CREB1* was expressed at higher levels in the FP_HBB group than in the FP_LBB and FP_LB+ groups, but there was no difference in the LP_HBB, LP_LBB, and LP_LB+ groups (Figure 2B,C). To investigate the role of *CREB1* in regulating female reproductive processes, we designed specific short interfering RNAs that target *CREB1* and constructed an overexpression vector. We discovered that *CREB1* siRNAs effectively suppressed its expression, and the overexpression vector significantly increased its expression (Figure 2D–G). Interference with *CREB1* significantly reduced the mRNA expression levels of *PTGS2*, *PRL*, *LIF*, *HOXA11*, *OPN*, and *VEGF*, whereas overexpression of *CREB1* significantly increased the mRNA expression levels of *PTGS2*, *LIF*, *MMP9*, and *BMP2* (Figure 2H,I). In addition, the results of immunohistochemistry (IHC) showed that the CREB1 protein was present in all reproductive tissues of ewes, including the hypothalamus, pituitary, oviduct, ovary, and uterus (Figure 2J). The CREB1 positive signals have been widely observed in the hypothalamus and pituitary tissues. In the oviduct, the CREB1 protein was present in the glandular epithelium, luminal epithelium, and cilia. In the follicles, no detectable presence of CREB1 was observed in oocytes from the primordial follicle stage onwards, oocyte cytoplasm, and granulosa cells around primary and secondary follicles, whereas a weakly CREB1 positive signal was detected in granulosa cells and theca cells of tertiary follicles. In the uterus, CREB1 was widely expressed in the endometrial luminal epithelium, glandular epithelium, endometrial stroma, and myometrium. Moreover, CREB1 was observed to have the highest expression in the hypothalamus, followed by the uterus, and the lowest expression in the oviduct (Figure 2K). Finally, the cellular localization of CREB1 was detected in ESCs by using IF, and the results displayed that the nucleus predominantly contained the majority of it, whereas a minor presence was observed in the cytoplasm (Figure 2L).

### 2.3. CREB1 Promotes Sheep Endometrial Stromal Cell Proliferation

*CREB1* interference significantly suppressed cell proliferation, as evidenced by the results obtained from the CCK8 and EdU assays, whereas *CREB1* overexpression promoted cell proliferation (Figure 3A–C). Compared to the controls, cell cycle research revealed that *CREB1* interference increased the proportion of cells in the G0/G1 phase, decreased the proportion of cells in the S phase, and weakly reduced the proportion of cells in the G2/M phase (Figure 3D). *CREB1* overexpression had contrasting effects on the number of cells in the G0/G1 and S phase (Figure 3F). However, consistent with *CREB1* interference, *CREB1* overexpression also decreased the proportion of cells in the G2/M phase (Figure 3F). The results of qRT-PCR revealed that *CREB1* interference resulted in a significant reduction in the mRNA expression levels of *PCNA*, *CCND1*, *CCND2*, *CCND3*, and *CCNA1*, except for the significant upregulation of *CDK6*, whereas *CREB1* overexpression increased the mRNA expression of these genes, except for the significant downregulation of *PCNA* (Figure 3E,G). The Western blotting analysis revealed that *CREB1* interference reduced the protein expression levels of PCNA, CDK2, CDK4, CDK6, and CCND1 compared to the negative group, whereas *CREB1* overexpression promoted the protein expression of these proteins except CDK6 (Figure 3H,I).

### 2.4. CREB1 Regulates Sheep Endometrial Stromal Cell Apoptosis and Autophagy

Compared with the controls, *CREB1* interference significantly promoted the apoptosis rate, whereas *CREB1* overexpression had the opposite effect (Figure 4A,B). The results of qRT-PCR revealed that *CREB1* interference increased the mRNA and protein expression of BAX and BCL2 but had no effect on the ratio of BAX/BCL2 (Figure 4C,G,H). Moreover, the mRNA expression of *CASP3* was also upregulated after *CREB1* interference (Figure 4C). *CREB1* overexpression decreased the mRNA levels of *BAX* and *BAX*/*BCL2* ratio while increasing the mRNA expression of *BCL2* (Figure 4D). The Western blotting analysis revealed that *CREB1* overexpression led to an increase in BCL2 protein levels and a reduction in the BAX/BCL2 ratio (Figure 4I,J). Autophagy and apoptosis are related at multiple levels and are important for maintaining cell homeostasis. Their interactions can be summarized into three types: cooperative, antagonistic, and facilitating. In this study, several important genes/proteins involved in the regulation of autophagy were also examined. The qRT-PCR analysis revealed that the mRNA levels of *LC3B*, *Beclin-1*, *ATG5*, and *ATG7* were significantly upregulated after interference with *CREB1*, whereas the mRNA levels of *LC3A*, *LC3B*, and *Beclin-1* were significantly downregulated, and the mRNA expression of *P62* was remarkably upregulated after overexpression of *CREB1* (Figure 4E,F). Further Western blot analysis revealed that the ratio of LC3-II/LC3-I, Beclin-1, and ATG7 was significantly upregulated, and the protein expression of P62 was moderately downregulated after interference with *CREB1* (Figure 4G,H). On the contrary, the protein levels of Beclin-1 and ATG7 were significantly downregulated, and the protein levels of P62 were largely upregulated after overexpression of *CREB1* (Figure 4I,J).

### 2.5. CREB1 Is a Direct Target of miR-134-5p

After analyzing the online databases (TargetScan, microT, ENCORI, and miRDB), miR-134-5p was identified as a potential miRNA that targets the *CREB1* gene (Figure 5A). Subsequently, we performed an analysis to investigate the association between miR-134-5p and *CREB1* expression levels. Our findings unveiled a significant negative correlation between them (Figure 5B). MiRanda and RNAhybrid analyses revealed a putative seed sequence match between miR-134-5p and the 3′UTR region of CREB1 (Figure 5C,D). To explore the potential targeted binding relationship between miR-134-5p and *CREB1*, the putative binding site of miR-134-5p within the 3’UTR region of *CREB1* was cloned into a luciferase reporter (Figure 5E). The results demonstrated a significant decrease in luciferase activities of the CREB1 wild-type reporter upon transfection with miR-134-5p mimics compared to the control reporter. Importantly, the restoration of luciferase activities was observed upon modification the sequence of *CREB1* (Figure 5F). Although the mRNA level of *CREB1* did not change, the protein level of CREB1 was markedly reduced after miR-134-5p upregulation in ESCs (Figure 5G,H). Repression of miR-134-5p, conversely, resulted in an increase in both the mRNA and protein levels of CREB1 (Figure 5G,H).

### 2.6. miR-134-5p Suppresses Sheep Endometrial Stromal Cell Proliferation

To investigate the regulatory role of miR-134-5p in female reproductive regulation, we overexpressed or inhibited miR-134-5p in sheep ESCs, and the expression of miR-134-5p was significantly upregulated or downregulated, respectively (Figure 6A,B). The overexpression of miR-134-5p increased the percentages of active and proliferating ESCs compared to control mimics, whereas miR-134-5p had the opposite effect (Figure 6C-F). Further research found that miR-134-5p overexpression increased the proportion of cells in the G0/G1 phase and decreased the proportion of cells in the S phase, with the converse changes observed upon miR-134-5p inhibition (Figure 6G,I). At the same time, miR-134-5p overexpression led to a significant decrease in the mRNA expression levels of *CDK4*, *CCND1*, *CCND2*, and *CCNA1* (Figure 6H). Inhibiting miR-134-5p significantly upregulated the mRNA expression of PCNA and CCNA1 (Figure 6J). The results of Western blotting revealed that overexpression of miR-134-5p significantly decreased the protein expression of PCNA, CDK2, CDK4, CDK6, and CCND1, whereas the inhibition of miR-134-5p increased the protein expression of CDK2 (Figure 6K–M).

### 2.7. miR-134-5p Regulates Sheep Endometrial Stromal Cell Apoptosis and Autophagy

Compared with the controls, miR-134-5p overexpression significantly increased the apoptosis rate, and miR-134-5p interference had the opposite effect at the same time (Figure 7A,C). The results of qRT-PCR revealed that miR-134-5p overexpression increased the mRNA expression of *BAX*, *BCL2*, *CASP3*, and *CASP9* but had no effect on the *BAX*/*BCL2* ratio, whereas miR-134-5p interference decreased the mRNA expression of *BAX*, *CASP9*, and the *BAX*/*BCL2* ratio (Figure 7B,D). The additional results of Western blotting revealed that miR-134-5p overexpression resulted in the downregulation of BCL2 protein expression and upregulation of the BAX/BCL2 ratio, whereas miR-134-5p interference showed the opposite results (Figure 7G–I). Then, we detected the expression levels of genes and proteins involved in cell autophagy. The results of qRT-PCR showed that miR-134-5p overexpression increased the mRNA expression of *LC3B*, *ATG5*, and *ATG7*, while miR-134-5p interference decreased the mRNA expression of *LC3A*, *LC3B*, *Beclin-1*, *ATG7*, and increased the mRNA expression of *P62* (Figure 7E,F). Further Western blotting analysis showed that miR-134-5p overexpression resulted in the upregulation of LC3-II, Beclin-1, ATG5, and ATG7 protein levels, an increased LC3-II/LC3-I ratio and a reduction in P62 protein levels (Figure 7G,H). Conversely, miR-134-5p interference showed an obviously opposite trend (Figure 7G,I).

### 2.8. CREB1 Rescues miR-134-5p-Induced Cell Cycle Arrest, Apoptosis, and Autophagy in Sheep Endometrial Stromal Cells

To further validate the involvement of miR-134-5p in regulating the positive effects of CREB1 in ESCs, pEX3-CREB1 was used to treat miR-134-5p-overexpressing ESCs. The results showed that the upregulation of CREB1 counteracted the decrease in cell viability and proliferation ability induced by miR-134-5p (Figure 8A–C). Furthermore, the upregulation of CREB1 also mitigated the miR-134-5p-induced G0/G1 phase arrest (Figure 8D,E). The overexpression of *CREB1*, as expected, attenuated cell apoptosis induced by transformation with miR-134-5p mimics (Figure 8F,G). Moreover, qRT-PCR and Western blotting assays confirmed our previous results (Figure 8H–J).

## 3. Discussion

Fecundity is one of the most important economic factors in sheep breeding and reproduction [9]. Ovulation rate and endometrial receptivity are two of the most crucial factors that influence fecundity. However, very few studies have been performed on the endometrial receptivity linked to ewe fecundity. In the present study, we found that in the luteal phase, high-fertility Hu sheep had a higher uterine coefficient, a richer uterine gland density, and a larger number of uterine caruncles than low-fertility Hu sheep. Ten genes associated with endometrial receptivity were considered in this study. In recent reports, the roles of CREB1 in mediating the development of endometriosis have been noteworthy [35,36,41]. Gain-of-function and loss-of-function assays revealed that CREB1 markedly affected the expression level of endometrial receptivity marker genes in ESCs, promoted endometrial stromal cell proliferation, and inhibited cell apoptosis and autophagy. Further mechanistic exploration revealed that *CREB1* expression was regulated by miR-134-5p in ESCs. Correspondingly, these data suggested that miR-134-5p-mediated CREB1 might participate in the formation of endometrial receptivity by regulating the function of ESCs.

The presence of uterine glands and their secretions play a crucial role in facilitating the establishment of uterine receptivity and the successful implantation of blastocysts [42]. Moreover, the endometrial microvessels and uterine volume are two critical indexes to evaluate endometrial receptivity [43,44]. Consistent with our findings in the follicular phase [40], we found that the uterine gland density and uterine coefficient of high-fertility Hu sheep were higher than that of low-fertility Hu sheep in our study, which is essential for enhancing the secretion ability of the endometrium, forming greater fetal capacity, and rich nutrition supply to maintain large litters. As reported, Lawrenz et al. demonstrated that the expression of the PLGF gene was higher in the endometrium of women with successful implantation compared to women who had a good endometrium but failed implantation [45]. In addition, it has been reported that human cytotrophoblast cells promoted endothelial cell survival and vascular remodeling by secreting PLGF, and its expression was firmly correlated with MVD in the evaluation of pathological angiogenesis [46]. As a result, the high-fertility Hu sheep may have greater angiogenesis, which is mainly due to the influence of the maternal placenta. To some extent, these data demonstrate that the endometrial receptivity of high-fertility Hu sheep was better than that of low-fertility Hu sheep.

Previous studies have reported several key pathways that are closely related to sheep reproductive performance, including the MAPK, Wnt, ECM-receptor interaction, cGMP-PKG, Calcium, and cAMP signaling pathways [40,47,48,49,50,51,52,53]. In our KEGG pathway analysis, we also screened these pathways. Then, a number of important genes that are closely related to fertility and uterine receptivity were focused on, including *BMP2*, *CREB1*, *ESR2*, *IGF2*, *PPP2R2A*, *PTGS2*, *SMAD1*, *SMAD2*, *STAT3*, and *WNT6*. Luo et al. reported that BMP2 played an important role in human endometrial remodeling by increasing the expression of IGFBP3 and MMP2 [54]. *Smad1/5* cKO mice showed endometrial receptivity defects and perturbed embryo implantation, and *Smad2/3* cKO mice showed endometrial disorders, sterility, and uterine cancer [55,56]. In humans, studies on CREB1 and ESR2 showed that they indirectly promoted the development of endometriosis [36,57,58]. A recent study indicated that the proper preparation of the uterus for implantation requires the sequential activation of uterine epithelial IGF1R by stromal IGF1 and embryonic IGF2 [59]. PPP2R2A has been demonstrated to promote implantation of embryos in vitro and in vivo, respectively, via *Ppp2r2a* knockout mice and function gain and loss assays of *PPP2R2A* in sheep ESCs [60,61]. PTGS2 was reported to be more abundant in repeat breeder cows (RBC) than in non-RBC and is involved in the regulation of some protein markers in the in vitro endometrial receptivity of endometrial epithelial cells (EEC) in dairy goats [62,63]. Hiraoka et al. reported that the collaborative regulation of uterine receptivity and embryo attachment involves distinct pathways utilized by epithelial and stromal STAT3, resulting in synergistic effects [64]. WNT6 demonstrated a higher abundance in polytocous Hu sheep than in monotocous Hu sheep and promoted uterine gland organogenesis by regulating the cell process of endometrial epithelial cells [40,65]. These events suggest that the above genes may contribute to the differences in litter size by regulating the establishment of endometrial receptivity in Hu sheep.

Compared with normal endometrial tissues, CREB1 has a higher expression level in ectopic endometrial tissues, and it affects endometrial cell motility, growth, and invasion under endometriosis (EMS) [35,36]. In addition, patients with clinically diagnosed EMS typically have reduced endometrial receptivity, expressed as abnormal expression of endometrial genes, sex hormone receptors, and cell adhesion molecules [37]. In our experiments, a significantly higher expression of *CREB1* was observed during the follicular phase in comparison to the luteal phase. Combined with previous research, we suspected that lower expression of CREB1 is more likely to maintain good uterine receptibility in the luteal phase. However, interference with *CREB1* significantly downregulated the expression level of several key genes regulating endometrial receptivity, including *PTGS2*, *PRL*, *LIF*, *HOXA11*, *OPN*, *VEGF*, *MMP9*, and *BMP2* [66,67,68,69,70,71,72,73,74], which seems to be contrary to previous assumptions. The reason may be that the establishment of endometrial receptivity is itself a very complex process, and the downstream molecule changes caused by different phases are not consistent. In cellular experiments, we observed that silencing of *CREB1* suppressed the proliferation of ESCs by blocking the G1 transition to the S phase. Available evidence indicates that *CREB1* activation positively regulates cell survival by regulating some cyclins, which were observed in our present studies, such as *CDK2*, *CDK4*, *CDK6*, and *CCND1* [75,76,77]. Moreover, our results showed that CREB1 affected the apoptosis in ESCs mainly via the regulation of the BAX/BCL2 ratio, which directly determines the openness of various channels in the mitochondrial outer membrane [78]. Consistent with our observations, Zhang et al. found that knockdown of *CREB1* promoted apoptosis in mouse granulosa cells in combination with a decrease in *BCL2* and an increase in *BAX* [77]. In a study of therapeutics targeting cancers, mTOR activation triggered CREB1 phosphorylation and accumulation, and inhibiting *CREB1* led to a decrease in *ATG7*, which is a key factor in autophagy regulation [79]. In the present study, the opposite trend of *ATG7* was observed in our results after *CREB1* knockdown. Several studies have reported inconsistent effects of *CREB1* on cancer cells and normal cells, which may explain the discrepancy between our results and those of other researchers. However, our detection methods for apoptosis and autophagy are not comprehensive enough due to the specificity of the sheep species, and the application of model animals will provide the possibility to further explore the deep molecular mechanisms of CREB1. Although we have some limitations in the detection method, our present results are sufficient to conclude that *CREB1* positively affected the function of ESCs by promoting the G1 to S phase transition and inhibiting apoptosis and autophagy.

The miRNA-meditated regulation of mRNAs has a significant influence on endometrial function. The inhibition of NDRG1 by miR-182-5p in humans, for instance, leads to the disruption of embryo implantation via the activation of the NF-κB/ZEB1/E-cadherin pathway [80]. Additionally, miR-182 aids in receptive endometrium development in dairy goats by downregulating PTN expression [81]. The regulatory association between miR-134-5p and CREB1 was elucidated via the integration of bioinformatics prediction, qRT-PCR analysis, Western blotting techniques, and luciferase reporter assays. The miR-134-5p has also been identified as a potential circulating biomarker for endometriosis [82], and the relationship between CREB1 and miR-134-5p has been extensively documented in the existing literature. Yang et al. showed that miR-134-5p directly targets *CREB1* to reduce infarct-induced cardiomyocyte apoptosis [83]. Consistent with these observations, our data suggested that the upregulation of CREB1 mitigated the detrimental effects induced by miR-134-5p overexpression in ESCs. Therefore, miR-134-5p was involved in regulating the function of ESCs by modulating CREB1 expression. Currently, miRNA is mainly regulated at four levels: epigenetic, transcriptional, post-transcriptional, and degradation. At the transcriptional level, the transcription factor (TF)-miRNA interaction network has been extensively studied. For example, Okada et al. found that the transcription factor p53 can negatively regulate miR-34a expression. Additionally, the presence of miR-34a can suppress the expression of the p53 negative regulator HDM4, thus forming a positive feedback pathway, which is conducive to the formation of tumors [84]. Therefore, miR-134-5p may also be regulated by CREB1, and the two form positive and negative feedback loops to work together.

## 4. Conclusions

In conclusion, we have shown that it was relatively easier to establish excellent endometrial receptivity in high-fertility Hu sheep compared to low-fertility Hu sheep. The results of transcriptome sequencing suggested some vital genes were involved in the establishment of endometrial receptivity. Importantly, the preliminary investigation suggests that the influence of CREB1 on the fertility of ewes represents a promising and hitherto unexplored determinant. It was regulated by miR-134-5p and may contribute to the establishment of endometrial receptivity by modulating the function of ESCs. Notwithstanding the positive outcomes observed, it is imperative to acknowledge certain limitations inherent in this study. For the in vitro experiments, a single cell type was exclusively utilized, disregarding potential intercellular interactions or paracrine signaling mechanisms. In addition, future in vivo validation is needed to confirm these conclusions. Despite its limitations, the present study provides crucial support for comprehending the regulatory mechanism of CREB1 mediating endometrial receptivity and provides an effective platform for studying the regulation mechanism of fertility in sheep.

## 5. Materials and Methods

### 5.1. Animals, Tissue Collection

Based on the results of *BMPR-IB* polymorphism genotyping, a total of sixteen pluriparous ewes with *FecB^BB^* genotypes and four pluriparous *FecB^B+^* were selected from Jiangsu Xilaiyuan Ecological Agriculture Co., Ltd. (Taizhou, China). The selected ewes were approximately three years old and were similar in weight. Ewes were divided into LP_HBB (LP, luteal phase; high prolificacy, each record of litter size = 3; *n* = 4; body weight of pre-slaughter: [47.98 ± 1.24] kg), LP_LBB/LP_LB+ (low prolificacy, each record of litter size = 1; *n* = 4; body weight of pre-slaughter: [52.05 ± 2.95] kg/[53.03 ± 4.69] kg), FP_HBB (FP, follicular phase; high prolificacy, each record of litter size = 3; *n* = 4; body weight of pre-slaughter: [47.43 ± 2.64] kg), FP_LBB (low prolificacy, each record of litter size = 1; *n* = 4; body weight of pre-slaughter: [51.05 ± 4.80] kg) groups based on identical litter size numbers of three records. The sheep utilized in this investigation were provided with unrestricted access to water and food under natural lighting conditions. Female sheep received synchronized estrus treatment, which involved the initial insertion of a vaginal sponge for a duration of 11 days. The estrus status of the ewes was assessed daily through interactions with a ram. Finally, ewes were slaughtered separately within 12 h after the detection of estrus (follicular phase) and on the ninth day (luteal phase), and the lateral endometrium was opened longitudinally to collect the endometrium from one side of the uterine horn. Female reproductive organs (hypothalamus, pituitary, uterus, oviduct, and ovary) were also collected for tissue expression pattern analysis. All samples were immediately stored at −80 °C. The other uterine horn was immersed in a 4% paraformaldehyde (PFA) solution for 24 h and subsequently embedded in paraffin for further analysis. In addition, intensive blood collection was performed before the sheep were slaughtered, as previously described [85]. The results of the BMPR-IB polymorphism genotyping are shown in Appendix A.

### 5.2. Morphological Analysis of Uterine Tissue

The uterus coefficient was determined by weighing the intact uterine organ and by dividing it by the total body weight of each ewe.

To quantify uterine gland density, 6 μm sections were subjected to staining using hematoxylin and eosin (H&E) following established protocols. For each sample, 5 randomly selected fields were imaged at 20× magnification using a Nikon microscope (ECLIPSE Ti, Tokyo, Japan). Gland counts per 20× field were determined via manual scoring using ImageJ software in a blinded fashion.

For microvessel density (MVD) quantification, adjacent serial sections were immunostained with a primary CD34 antibody using a standard streptavidin-biotin horseradish peroxidase method. Five areas were randomly selected under the Nikon microscope (ECLIPSE Ti, Tokyo, Japan). Any brown endothelial cell or cell mass that could be easily distinguished from the adjacent tissues was defined as a single microvessel [86].

Endometrial DGI was quantified by examining the entire uterine section using a Nikon microscope (ECLIPSE Ti, Tokyo, Japan). The glandular epithelium was distinctly cavated and was regarded as DGI, and the number of DGI in the whole uterus of each sheep in the luteal phase was counted (see Appendix A for the schematic diagram).

The schematic diagrams of the LE thickness and myometrial thickness analysis are shown in Appendix A, and ten positions were evenly selected for analysis.

Analysis of endometrium thickness: To assess the thickness of sheep endometria, the endometrial thickness (ET) was calculated using the formula ET = 2A/(P1 + P2) [87,88]. The calculated area (A) is the total endometrial area on a uterine cross-section. The calculated inner perimeter (P1) was measured by tracing the lumen and the endometrium–myometrium interface. The outer perimeter (P2) was determined by tracing this interface (as shown in Appendix A). All the measurements were determined using the ImageJ software (National Institute of Health, Bethesda, MD, USA).

### 5.3. RNA-Seq and Bioinformatics Analysis

The total RNA was extracted from twelve Hu sheep endometrial tissues in the luteal phase using the TRIzol reagent (Thermo Scientific, Waltham, MA, USA). The RNA quantity and purity were assessed using NanoDrop ND-1000 (NanoDrop, Wilmington, DE, USA), and only RNA samples with RIN (RNA integrity) > 7 were used for sequencing. To prepare the samples of RNA-seq, we utilized a Ribo-Zero™ rRNA Removal Kit (Illumina, San Diego, CA, USA) to deplete ribosomal RNA from ~5 µg of total RNA per sample. The NEB Next^®^ Ultra Directional RNA LibraryPrep Kit for Illumina^®^ (NEB, Ipswich, MA, USA) was employed to construct the strand-specific RNA-seq libraries. Then, paired-ended sequencing was performed using an Illumina NovaSeq 6000 Sequencing System to generate paired-end 150 bp reads, with a sequencing depth of at least 12 million reads per sample (LC-Bio Technology Co., Ltd., Hangzhou, China). Raw data from nine Hu sheep endometrial tissues in the follicular phase were derived from previous laboratory sequencing data [40]. The generated data underwent quality assessment using the Fast QC tool (version 0.10.1) and were aligned to the reference genome for clean reads data. Differentially expressed genes (DEGs) were identified using at least 1.5 fold changes and *p*-value < 0.05. Volcano plots, the enrichment analysis of Gene Ontology (GO), and the Kyoto Encyclopedia of Genes and Genomes (KEGG) can be carried out using the OmicStudio tools (http://www.omicstudio.cn/tools, accessed on 3 July 2023). Volcano plots were generated from significantly regulated genes. The enrichment of relevant GO terms and KEGG pathways was considered significantly enriched when their adjusted *p*-value were below 0.05.

### 5.4. RNA Preparation and Real-Time Quantitative PCR (qRT-PCR)

The total RNA was extracted in accordance with the manufacturer’s Trizol protocol (Invitrogen, Waltham, CA, USA). RNA concentration and purity were assessed via spectrophotometry (A260/280 > 1.8). Total RNA (1 μg) was reverse-transcripted using the HiScript II 1st Strand cDNA Synthesis Kit (+gDNA wiper) (37 °C for 15 min, 85 °C for 5 s) or the miRNA 1st Strand cDNA Synthesis Kit (by stem-loop) (25 °C for 5 min, and 50 °C for 15 min, followed by 85 °C for 5 min) (Vazyme, Nanjing, China). Genomic DNA was removed prior to the first strand cDNA synthesis (42 °C for 2 min). The qRT-PCR was performed on a QuantStudio™ 7 Real-Time PCR system (Applied Biosystems, Foster, CA, USA) using the PerfectStart^®^ Green qPCR SuperMix (+Dye II) (TransGen Biotech, Beijing, China) (94 °C for 30 s, and 40 cycles of 94 °C for 5 s, 60 °C for 30 s, followed by 95 °C for 15 s, 60 °C for 1 min, and 95 °C for 15 s). The expression levels of candidate mRNAs and miRNAs were initially normalized to those of ACTB and U6, respectively, and the 2^−ΔΔCt^ method was employed for calculating the expression levels. The PCR specificity was verified by melt curve analysis. The primers employed in this study can be found in Appendix A.

### 5.5. Western Blotting

Total protein was extracted from the ESCs using RIPA lysis buffer (Invitrogen, Waltham, CA, USA) containing protease and phosphatase inhibitor cocktail (Beyotime, Haimen, China) and quantified using a BCA kit (Beyotime, Haimen, China) as per the manufacturer’s protocol; 30 µg of total protein was resolved using 12% sodium dodecyl sulfate-polyacrylamide gels and subsequently transferred in Tris-glycine buffer at a voltage of 100 V for 1 h. Membranes were blocked by incubating them in a 5% non-fat milk solution in Tris-HCL buffer with Tween-20 (TBST) for 1 h at room temperature. Subsequently, the primary antibodies were incubated overnight at 4 °C, followed by the application of the secondary antibodies at room temperature for 1 h (the antibody dilutions adhered to the specifications provided in Appendix A). The chemiluminescence detection kit (Biosharp, Guangzhou, China) was utilized for protein band visualization with an electroluminescence detection system (Bio-RAD, Hercules, CA, USA). ImageJ software (National Institute of Health, Bethesda, MD, USA) was employed to quantify the intensity of the protein bands, and Tubulin was used for normalizing the target proteins.

### 5.6. Immunohistochemistry (IHC)

The sections of reproductive axis tissue (hypothalamus, pituitary, oviduct, ovary, and uterus) were obtained from paraffin tissue blocks previously embedded in the laboratory. Consecutive sections containing the largest hypothalamic area were selected according to the ovine brain atlas [89]. The IHC analysis was performed using established methodologies as previously described [90]. Briefly, the wax was removed from each section by employing xylene and a series of gradually concentrated ethanol solutions. To quench the endogenous peroxidase activity, a 10-minute incubation was performed in methanol containing 3% hydrogen peroxide. The procedure for antigen retrieval involved exposing the tissue sections to a 5-minute microwave treatment in a citrate buffer solution. To minimize non-specific binding sites, we employed a 5% solution of bovine serum albumin (BSA) for blocking purposes. Subsequently, the tissue sections were incubated overnight at 4 °C with polyclonal rabbit antibodies against CD34/CREB1 (as specified in Appendix A). After the incubation period, the sections were washed three times using phosphate-buffered saline (PBS) and then exposed to the appropriate secondary antibody (as listed in Appendix A) at room temperature for 1 h. Relevant experiments were conducted by substituting the primary antibody with PBS as controls. Sections were developed with 3–3′-diaminobenzidine (DAB) (Boster, Guangzhou, China) for 5 min. After rinsing the sections with distilled water, a counterstaining procedure was performed using hematoxylin (Boster, Guangzhou, China). The sections were then air-dried and sealed with neutral resin. Imaging was performed using a Pannoramic MIDI digital sliDE scanner (3DHISTECH) at 20× magnification, and images were analyzed using SlideViewer software (Version 2.5).

### 5.7. Enzyme-Linked Immunosorbent Assay (ELISA)

The serum levels of E2, P4, PRL, and PLGF were measured using commercial ELISA kits according to the manufacturer’s instructions (Kmaels Co., Ltd., Shanghai, China). Absorbance at 450 nm was used to quantify hormone levels from the standard curves. The intra- and inter-assay CVs were 10% and 15%, respectively. The limits of detection were 1.0 pg/mL (E2), 0.1 ng/mL (P4), 1.0 ng/mL (PRL) and 1.0 pg/mL (PLGF).

### 5.8. Cell Culture and Treatment

The isolation, purification, and identification methods of ESCs follow our laboratory’s previous methods [60] (Appendix A). The stored cells were cultured in Dulbecco’s Modified Eagle Medium: F12 (DMEM/F12) (Thermofisher, Waltham, MA, USA), with 10% fetal bovine serum (FBS) (Gibco, Gland Island, NY, USA) and 1% penicillin-streptomycin solution (Gibco). Our laboratory’s 293T cells (human embryonic kidney cells) were cultured in a DMEM medium (Thermofisher) with 10% FBS and 1% penicillin-streptomycin. They were all cultured at 37 °C in an atmosphere of 5% CO_2_. The relative plasmid or oligonucleotide was transfected into ESCs and 293T cells using Lipofectamine 3000 (Invitrogen, Waltham, MA, USA) following the manufacturer’s protocol. Briefly, 4 × 10^5^ cells per well were seeded in 12-well plates and transfected at 70–80% confluency with plasmid DNA or small interfering RNAs (siRNAs). The transfection efficiency of Lipofectamine 3000 was optimized by transfecting a GFP expression plasmid and visualizing GFP expression by fluorescence microscopy 24 h post-transfection. In ESCs, transfection with 2 μL Lipofectamine 3000 and 1 μg plasmid DNA resulted in >50% GFP-positive cells. In 293T cells, >80% of cells expressed GFP under these conditions.

The siRNAs of *CREB1* (si-CREB1 1# and si-CREB1 2#), miR-134-5p mimics/inhibitors, and corresponding negative controls (NC) were purchased from GenePharma (Shanghai, China). The *CREB1* overexpression plasmid (pEX3-CREB1) and empty vector (pEX3-NC) were purchased from Tsingke Biotechnology Co., Ltd. (Beijing, China). The *CREB1* 3′UTR containing the miR-134-5p binding site was cloned downstream of Renilla in the pmirGLO luciferase reporter vector. Mutations in the miR-134-5p-binding sites in the *CREB1* 3′UTR sequence were generated using a Mutagenesis Kit (Vazyme, Nanjing, China). The sequences of siRNAs, mimics, or inhibitors are listed in Appendix A.

### 5.9. Immunofluorescence (IF)

The ESCs were washed three times with PBS and then fixed in 4% PFA for 15 min at room temperature. Subsequently, permeabilization was performed using 0.1% Triton X-100 for 30 min. Non-specific binding sites were blocked with 3% BSA. Then, the ESCs were incubated overnight at 4 °C with appropriate primary antibodies (as indicated in Appendix A). Negative controls were subjected to incubation with normal FBS instead of the primary antibody. After three washes with PBS, the cells were incubated at room temperature for 1 h with a secondary antibody conjugated to Alexa Fluor 594/488. Then, the cell nuclei were stained with 4, 6-diamidino-2-phenylindole (DAPI) (Beyotime, Haimen, China), and the cells were washed three times with PBS. Immunofluorescence was visualized using a Nikon Eclipse Ti inverted epifluorescence microscope using a 20× objective and NIS Elements imaging software (Version 4.10).

### 5.10. Cell Viability Assay

The assessment of cell viability was conducted using the Cell Counting Kit-8 (CCK-8) assay provided by Vazyme (Nanjing, China). The CCK-8 assay was performed by adding 10 μL of CCK8 reagent to individual wells of a 96-well plate, followed by incubation at 37 °C for a duration of 2 h. After the incubation period, the optical density at 450 nm was measured using a microplate reader (51119200, Thermo Scientific, Waltham, MA, USA).

### 5.11. Cell Proliferation

The assessment of cell proliferation was evaluated by 5-Ethynyl-2′-deoxyuridine (EdU) incorporation assay using an Alexa Fluor 555/488-ClickiT EdU Assay Kit (Keygen Biotech, Nanjing, China). Briefly, the ESCs (2 × 10^5^ cells per well) were seeded on glass coverslips in a 24-well plate and subjected to different transfection treatments. After 24 h, a working solution of EdU was added at a concentration of 50 μM. Subsequently, the cells underwent three washes with Dulbecco’s Phosphate-Buffered Saline (DPBS) and then fixed with 4% PFA for 30 min. A rinse with a glycine solution (2 mg/mL) was performed for 5 min. After removing the glycine solution, cells were permeabilized in DPBS containing 0.5% Triton X-100 for 20 min. Following three additional washes in DPBS, Hoechst 33342 dye was used to stain cell nuclei. Finally, an antifade medium was applied to cover the images before capturing them using a laser scanning confocal microscope (LSM700META, Carl Zeiss, Berlin, Germany) under identical conditions. ZEN software (Version 2.6, Carl Zeiss, Berlin, Germany) was used to save and export the pictures.

### 5.12. Cell Cycle Analysis

According to the manufacturer’s instructions, propidium iodide (PI) staining (Keygen Biotech, Nanjing, China) was used to label the ESCs for cell cycle analysis. The DNA content was evaluated using flow cytometry (Beckman Coulter, Brescia, CA, USA), and the distribution cycle was quantified utilizing FlowJo software (Version 10.6.2, TreeStar, Ashland, OR, USA).

### 5.13. Cell Apoptosis Analysis

Apoptotic cells were detected using Annexin V FITC/PI staining. Initially, the transfected ESCs were washed three times with ice-cold PBS and then resuspended in 100 µL of binding buffer. Annexin V-FITC (5 µL, 20 mg/mL) and PI (10 µL, 50 mg/mL) (BD Biosciences, Franklin Lakes, NJ, USA) were added and incubated for 15 min under ambient conditions. Subsequently, 400 µL of binding buffer was added, and the cells were analyzed via flow cytometry using a Becton Dickinso FACS Calibur (Ashland, OR, USA). The apoptosis rates were determined using the FlowJo software (TreeStar) by analyzing the cellular distribution in quadrants Q2 (late apoptotic) and Q3 (early apoptotic).

### 5.14. Luciferase Activity Assay

The 293T cells were cultured in 24-well plates at a seeding density of 2 × 10^5^ cells per well. Co-transfected was performed using Lipofectamine 3000 regeant (Invitrogen, Waltham, CA, USA) to introduce the NC/miR-134-5p mimics and the luciferase reporter vector containing the 3′-UTR region of *CREB1.* Renilla luciferase was used as an internal control. After a transfection period of 48 h, luciferase activities were quantified using the Dual-Luciferase Reporter Assay System (Vazyme, Nanjing, China) according to the manufacturer’s operating instructions.

### 5.15. Statistical Analysis

At least three biological replicates were used in each experiment. Prior to analysis, data normality was confirmed using the Kolmogorov–Smirnov goodness-of-fit test. Student’s *t*-test was employed for two groups, while one-way analysis of variance (ANOVA) was used for multiple groups. If data violated assumptions of homoscedasticity, nonparametric Mann–Whitney U test or Kruskal–Wallis H test with Dunn’s post hoc test were used instead. Statistical analyses were conducted using SPSS 23.0 (IBM, Armonk, NY, USA) and GraphPad Prism 8.0 (Inc., San Diego, CA, USA). The data were presented as the mean ± standard error of the mean (SEM), which was calculated based on at least three independent experiments. Differences with *p*-value < 0.05 were considered statistically significant, whereas differences with *p*-value < 0.01 were deemed markedly statistically significant.

## Figures and Tables

**Figure 1 cells-12-02554-f001:**
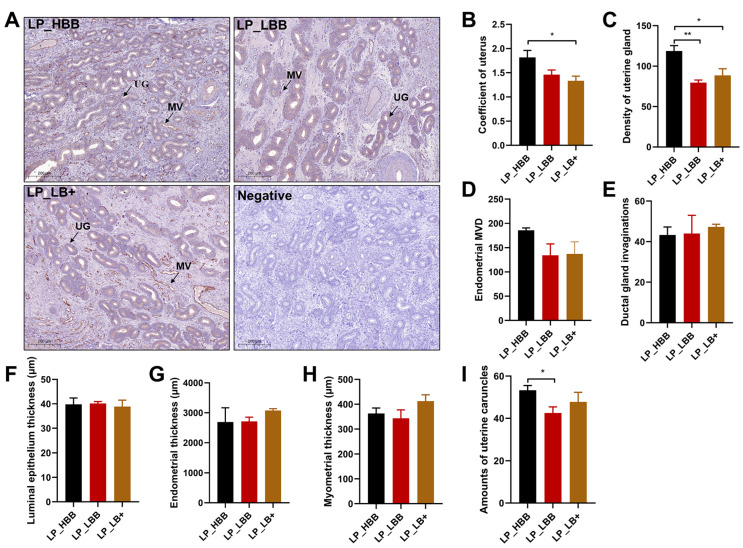
Analysis of uterine morphology of Hu sheep in the luteal phase. (**A**) The immunohistochemical analysis involved the localization of CD34 antibodies in endometrial microvessels to determine microvessel density (MVD). LP_HBB, high prolificacy group carrying the *FecB^BB^* genotype; LP_LBB, low prolificacy group carrying the *FecB^BB^* genotype; LP_LB+, *FecB^B+^*, low prolificacy group carrying the *FecB^B+^* genotype; LP, luteal phase; NC, negative control; UG, uterine glands; MV, microvascular; scale bar = 200 μm. (**B**) Analysis of the uterine coefficient. (**C**) Analysis of endometrial gland density. (**D**) Analysis of endometrial microvascular D (MVD). (**E**) Analysis of the endometrial ductal gland invaginations (DGI). (**F**) Analysis of the luminal epithelium (LE) thickness. (**G**) Analysis of the endometrial thickness. (**H**) Analysis of the myometrial thickness. (**I**) Number of uterine caruncles. Results are expressed as the mean ± SEM; * *p* < 0.05; ** *p* < 0.01.

**Figure 2 cells-12-02554-f002:**
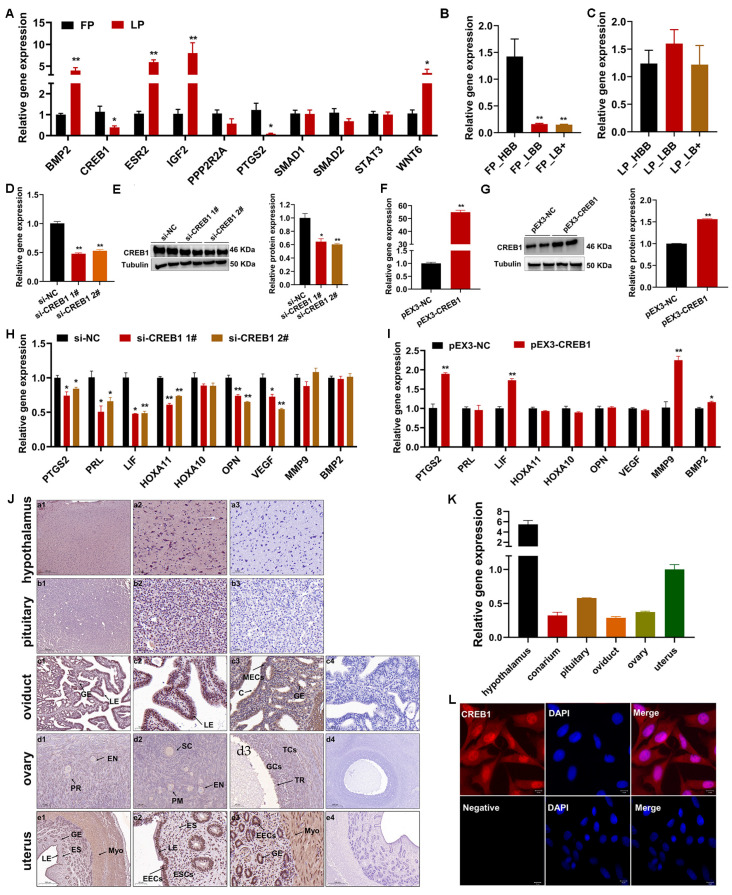
Characteristics of CREB1 during the estrous cycle of sheep. (**A**) The qRT-PCR detected the expression of some key node genes in the endometrium of Hu sheep with different fecundity. (**B**) The mRNA abundance of the *CREB1* gene in the uterus of the follicular phase of Hu sheep with different fertility potential. (**C**) The mRNA abundance of the *CREB1* gene in the uterus of the luteal phase of Hu sheep with different fertility potential. (**D**,**E**) The qRT-PCR and Western blotting measured the expression of CREB1 and Tubulin in ESCs transfected with the control or siRNAs. (**F**,**G**) The qRT-PCR and Western blotting measured the expression of CREB1 and Tubulin in ESCs transfected with the vehicle control or *CREB1* overexpression plasmid. (**H**,**I**) The effect of CREB1 on endometrial receptivity marker genes. (**J**) CREB1 staining in the hypothalamus, pituitary, oviduct, ovary, and uterus tissues. (**a3**,**b3**,**c4**,**d4**,**e4**) Negative control. The positive signals appeared brown, and the counterstained background appeared blue. GE: glandular epithelium; LE: luminal epithelium; C: cilia; MECs: mucosal epithelial cells; EN: oocyte nest; PM: primordial follicle; PR: primary follicle; SC: secondary follicle; TR: tertiary follicle; GCs, granulosa cells; TCs, theca cells; ES: endometrial stroma; Myo: myometrium; EECs: endometrial epithelial cells; ESCs: endometrial stromal cells. (**a1**,**e1**,**e4**), scale bar = 500 µm; (**c1**,**d4**), scale bar= 200 µm; (**a2**,**a3**,**b1**,**b2**,**c2**–**c4**,**d1**–**d3**,**e2**,**e3**), scale bar = 50 µm. (**K**) The mRNA abundance of the *CREB1* gene in the hypothalamus, pituitary, oviduct, ovary, and uterus tissues. (**L**) Localization of CREB1 (red) in sheep ESCs. The cell nuclei were stained with DAPI (blue). Scale bar = 10 µm. Results are expressed as the mean ± SEM; * *p* < 0.05, ** *p* < 0.01.

**Figure 3 cells-12-02554-f003:**
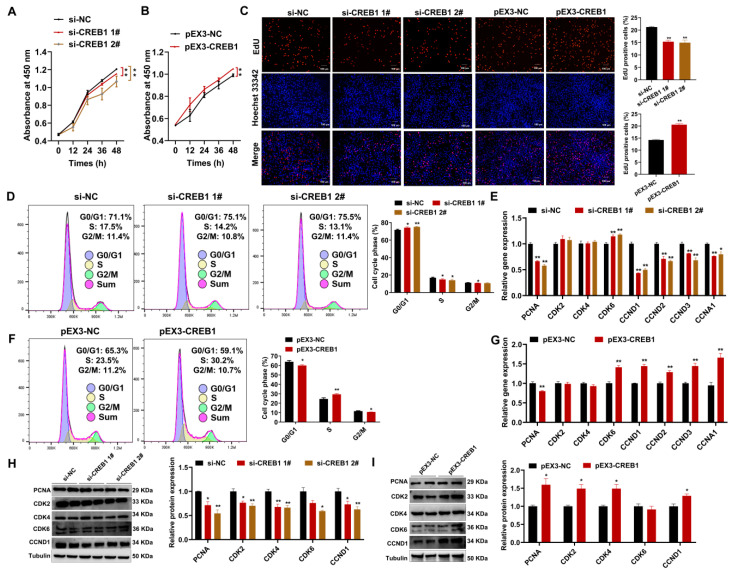
CREB1 promoted the proliferation of sheep endometrial stromal cell (ESC). (**A**) CCK8 assay detected the viability of sheep ESCs after transfection with control or siRNAs. (**B**) CCK8 assay detected the viability of sheep ESCs after transfection with vehicle control or *CREB1* overexpression plasmid. (**C**) EdU assay detected the number of positive EdU staining ESCs (red) after transfection with control or siRNAs and vehicle control or *CREB1* overexpression plasmid. The cell nucleus was stained with Hoechst33342, scale bar = 100 μm. (**D**) Flow cytometry analysis of cell cycle alterations in sheep ESCs after transfection with control or siRNAs. (**E**) The relative expression of cell cycle-related genes in sheep ESCs was assessed using qRT-PCR after transfection with control or siRNAs. (**F**) Flow cytometry analysis of cell cycle alterations in sheep ESCs after transfection with vehicle control or *CREB1* overexpression plasmid. (**G**) The relative expression of cell cycle-related genes in sheep ESCs was evaluated by qRT-PCR after transfection with vehicle control or *CREB1* overexpression plasmid. (**H**) The relative expression of cell cycle-related proteins in sheep ESCs was evaluated by Western blotting after transfection with control or siRNAs. (**I**) The relative expression of cell cycle-related proteins in sheep ESCs was evaluated by Western blotting after transfection with vehicle control or *CREB1* overexpression plasmid. Results are expressed as the mean ± SEM; * *p* < 0.05; ** *p* < 0.01.

**Figure 4 cells-12-02554-f004:**
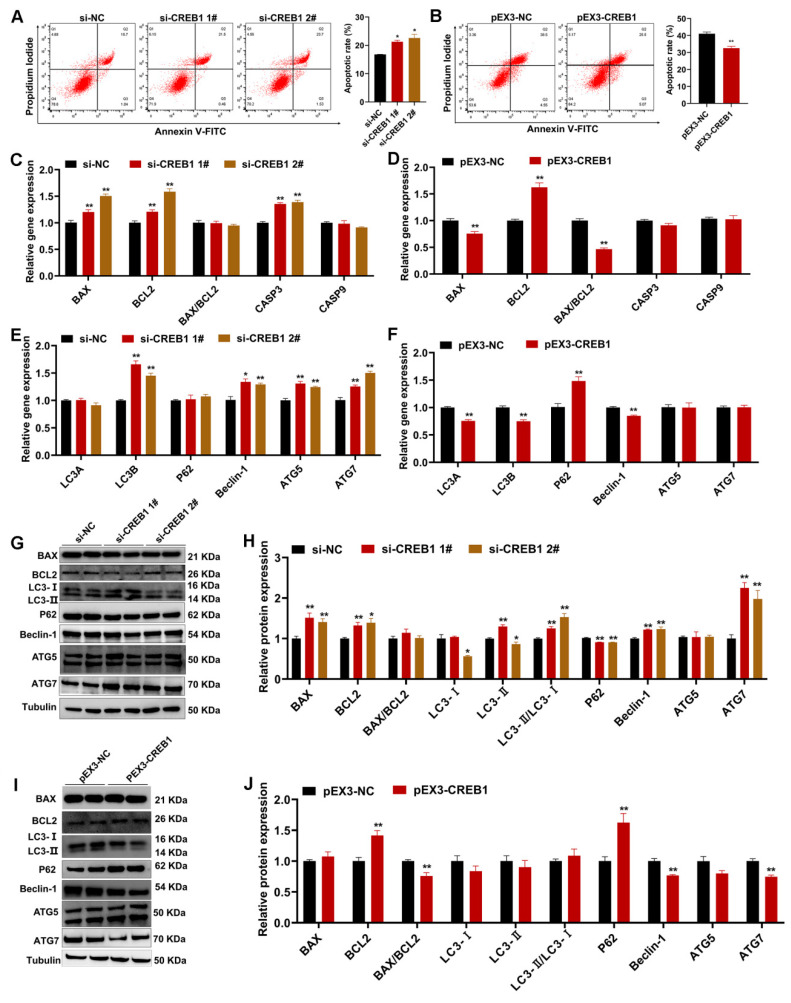
CREB1 regulated apoptosis and autophagy in sheep endometrial stromal cell (ESC). (**A**,**B**) Flow cytometry analysis of the apoptosis rate of sheep ESCs after transfection with control or siRNAs and vehicle control or *CREB1* overexpression plasmid. (**C**,**D**) The qRT-PCR detected the relative expression of cell apoptosis-related genes in sheep ESCs after transfection with control or siRNAs and vehicle control or *CREB1* overexpression plasmid. (**E**,**F**) The qRT-PCR detected the relative expression of cell autophagy-related genes in sheep ESCs after transfection with control or siRNAs and vehicle control or *CREB1* overexpression plasmid. (**G**,**H**) Western blotting measured the relative expression of cell apoptosis-related and autophagy-related proteins in sheep ESCs after transfection with control or siRNAs. (**I**,**J**) Western blotting measured the relative expression of cell apoptosis-related and autophagy-related proteins in sheep ESCs after transfection with the vehicle control or *CREB1* overexpression plasmid. Results are expressed as the mean ± SEM; * *p* < 0.05, ** *p* < 0.01.

**Figure 5 cells-12-02554-f005:**
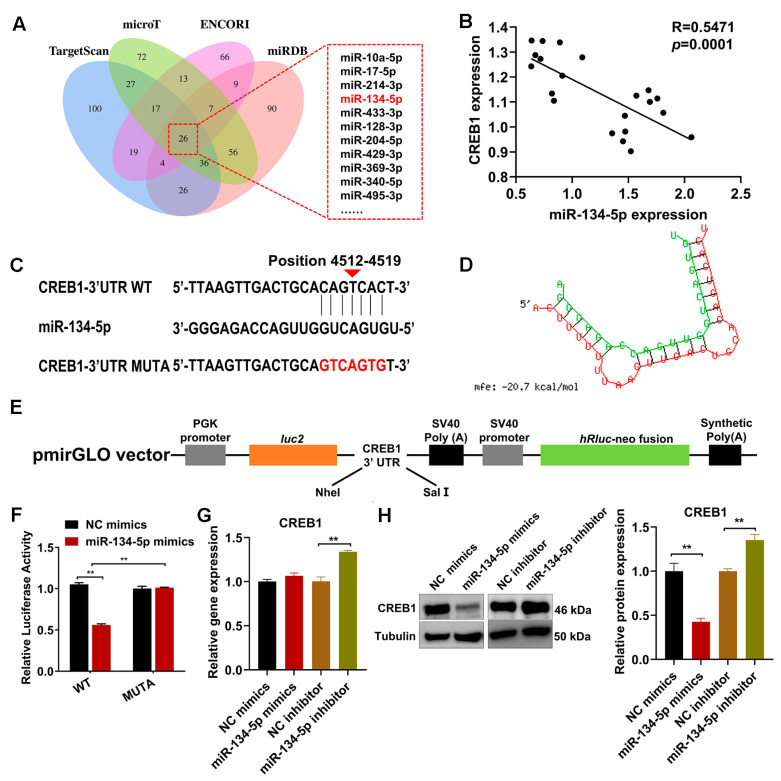
*CREB1* acted as a downstream effector of miR-134-5p. (**A**) Predicted miRNAs targeting *CREB1* according to four algorithms. (**B**) Statistical analysis was performed to investigate the correlation between miR-134-5p and *CREB1* in endometrial tissues. (**C**,**D**) The miR-134-5p binding site in the 3′UTR of *CREB1* was predicted using miRanda and RNAhybrid. (**E**) The wild-type (WT) and mutated (MUTA) *CREB1* 3′UTR were subcloned into the pmirGLO luciferase reporter vector. (**F**) The luciferase reporter assay showed direct binding between the miR-134-5p and the *CREB1* 3′UTR. (**G**,**H**) The relative expression of CREB1 in sheep ESCs was assessed using qRT-PCR and Western blotting following transfection with NC/miR-134-5p mimics and NC/miR-134-5p inhibitors. ** *p* < 0.01.

**Figure 6 cells-12-02554-f006:**
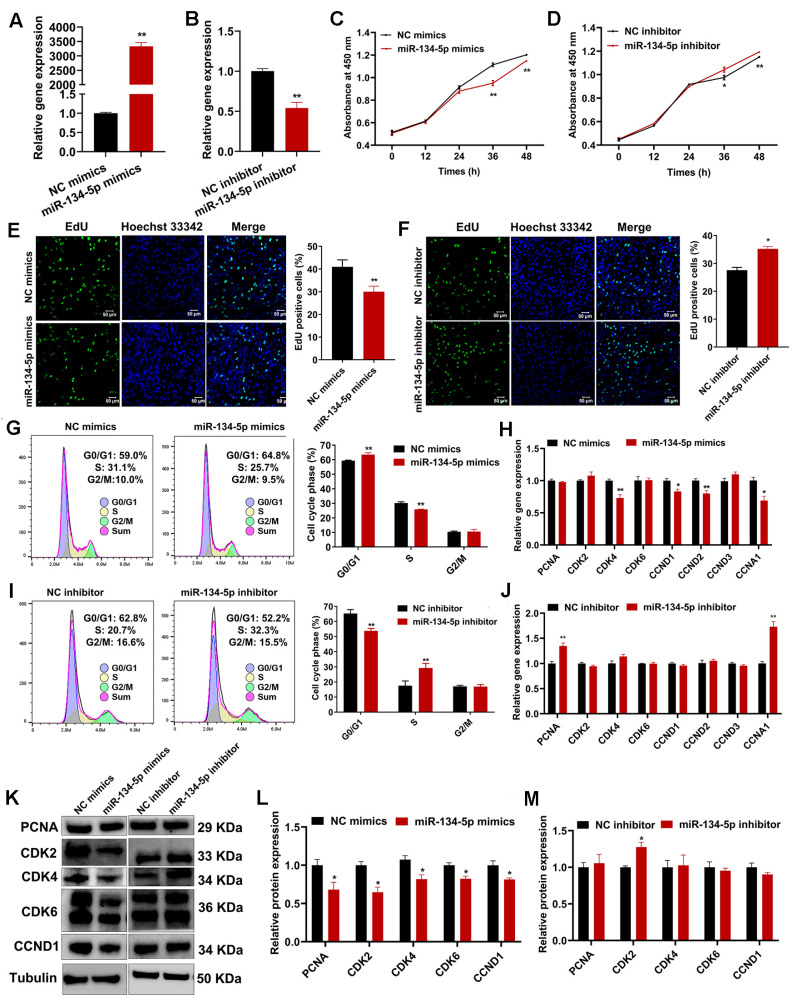
miR-134-5p suppressed sheep endometrial stromal cell (ESC) proliferation. (**A**,**B**) The qRT-PCR measured the relative expression of miR-134-5p in sheep ESCs after transfection with NC/miR-134-5p mimics and NC/miR-134-5p inhibitors. (**C**,**D**) CCK8 assays detected the viability of sheep ESCs after transfection with NC/miR-134-5p mimics and NC/miR-134-5p inhibitors. (**E**,**F**) EdU assays detected the number of positive EdU staining ESCs (green) after transfection with NC/miR-134-5p mimics and NC/miR-134-5p inhibitors. The cell nucleus was stained with Hoechst33342, scale bar = 50 μm. (**G**) Flow cytometry analysis of cell cycle alterations in sheep ESCs after transfection with NC/miR-134-5p mimics. (**H**) The qRT-PCR detected the relative expression of cell cycle-related genes in sheep ESCs after transfection with NC/miR-134-5p mimics. (**I**) Flow cytometry analysis of cell cycle alterations in sheep ESCs after transfection with NC/miR-134-5p inhibitor. (**J**) The qRT-PCR detected the relative expression of cell cycle-related genes in sheep ESCs after transfection with the NC/miR-134-5p inhibitor. (**K**–**M**) Western blotting measured the relative expression of cell cycle-related proteins in sheep ESCs after transfection with NC/miR-134-5p mimics and NC/miR-134-5p inhibitors. Results are expressed as the mean ± SEM; * *p* < 0.05; ** *p* < 0.01.

**Figure 7 cells-12-02554-f007:**
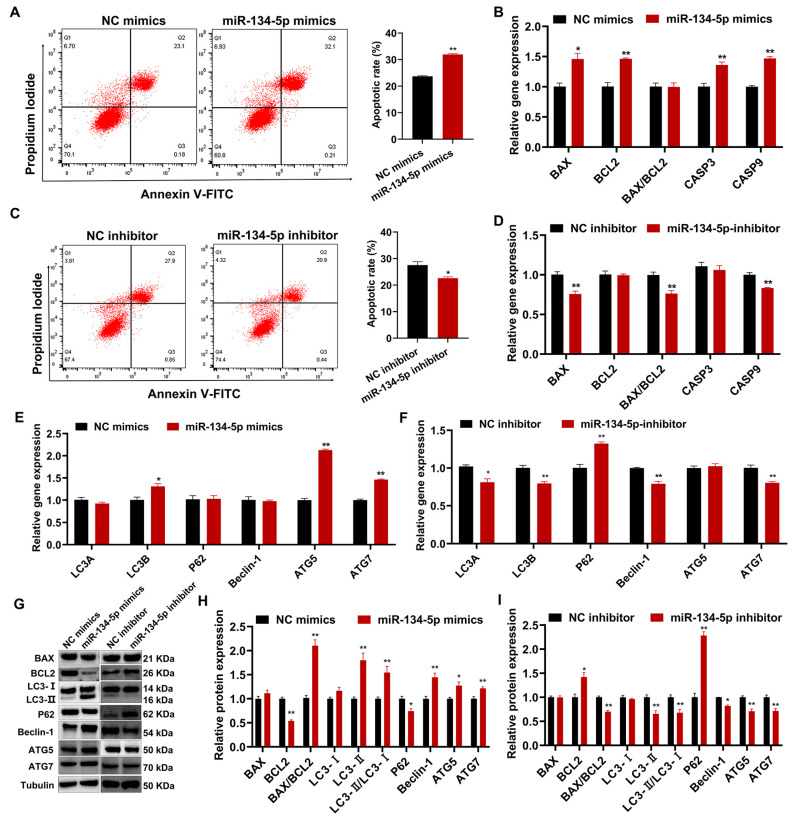
miR-134-5p regulated apoptosis and autophagy in sheep endometrial stromal cells (ESC). (**A**) Flow cytometry analysis of the apoptosis rate of sheep ESCs after transfection with NC/miR-134-5p mimics. (**B**) The relative expression of genes associated with cellular apoptosis was evaluated in sheep ESCs after transfection with NC/miR-134-5p mimics using qRT-PCR. (**C**) Flow cytometry analysis of the apoptosis rate of sheep ESCs after transfection with the NC/miR-134-5p inhibitor. (**D**) The relative expression of genes associated with cellular apoptosis was evaluated in sheep ESCs after transfection with NC/miR-134-5p inhibitor using qRT-PCR. (**E**,**F**) The qRT-PCR detected the relative expression of cell autophagy-related genes in sheep ESCs after transfection with NC/miR-134-5p mimics and NC/miR-134-5p inhibitors. (**G**–**I**) Western blotting measured the relative expression of cell apoptosis-related and autophagy-related proteins in sheep ESCs after transfection with NC/miR-134-5p mimics and NC/miR-134-5p inhibitors. Results are expressed as the mean ± SEM; * *p* < 0.05; ** *p* < 0.01.

**Figure 8 cells-12-02554-f008:**
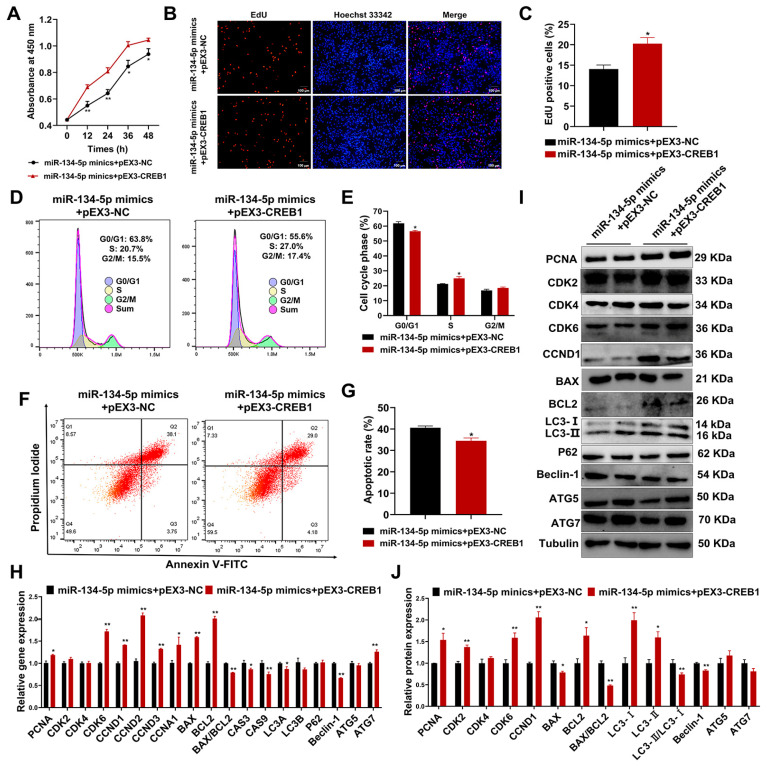
CREB1 rescued miR-134-5p-induced cell cycle arrest, apoptosis, and autophagy in sheep endometrial stromal cells. (**A**) CCK8 assays detected the viability of sheep ESCs after transfection with miR-134-5p mimics and vehicle control/*CREB1* overexpression plasmid. (**B**,**C**) EdU assays detected the number of positive EdU staining ESCs (red) after transfection with miR-134-5p mimics and vehicle control/*CREB1* overexpression plasmid. (**D**,**E**) Flow cytometry analysis of cell cycle alterations of sheep ESCs after transfection with miR-134-5p mimics and vehicle control/*CREB1* overexpression plasmid. (**F**,**G**) Flow cytometry analysis of the apoptosis rate of sheep ESCs after transfection with miR-134-5p mimics and vehicle control/*CREB1* overexpression plasmid. (**H**) The qRT-PCR detected the relative expression of cell cycle-related, apoptosis-related, and autophagy-related genes in sheep ESCs after transfection with miR-134-5p mimics and vehicle control/*CREB1* overexpression plasmid. (**I**,**J**) Western blotting detected the relative expression of cell cycle-related, apoptosis-related, and autophagy-related proteins in sheep ESCs after transfection with miR-134-5p mimics and vehicle control/*CREB1* overexpression plasmid. Results were expressed as mean ± SEM. * *p* < 0.05; ** *p* < 0.01.

## Data Availability

The data generated and analyzed during this study are available within this article and Appendix A.

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
