# Peer review of "CREB1 Is Involved in miR-134-5p-Mediated Endometrial Stromal Cell Proliferation, Apoptosis, and Autophagy"

_cells, 2023, doi:10.3390/cells12212554_

Round 1
Reviewer 1 Report
In this article, the author has discovered CREB1 promote endometrial stromal cell (ESC) growth and CREB1 can be regulated by miR-134-5p. This article is well-written, concise, and logically organized. Providing more detailed information could further facilitate the readers' understanding.
In Fig.4, autophagy and apoptosis will occur spontaneously during the growth process of healthy ESC? In the study of apoptosis and autophagy, inducers or starvation are commonly used to trigger apoptosis and autophagy. In addition, cleavage of PARP is the most common marker used for apoptosis. For autophagy, LC3B-GFP or other stanning for endogenous LC3 redistribution is necessary.
Author Response
In this article, the author has discovered CREB1 promote endometrial stromal cell (ESC) growth and CREB1 can be regulated by miR-134-5p. This article is well-written, concise, and logically organized. Providing more detailed information could further facilitate the readers' understanding.
In Fig.4, autophagy and apoptosis will occur spontaneously during the growth process of healthy ESC? In the study of apoptosis and autophagy, inducers or starvation are commonly used to trigger apoptosis and autophagy. In addition, cleavage of PARP is the most common marker used for apoptosis. For autophagy, LC3B-GFP or other stanning for endogenous LC3 redistribution is necessary.
We appreciate the reviewer raising this important point about the need for additional experiments to further characterize the mechanisms of apoptosis and autophagy in our system. Per the reviewer's recommendation, we first searched "sheep" and "PARP" through pubmed, and found that only one paper published in 2017 was reported in the literature with definite reports in recent ten years1. Then we purchased the corresponding antibody (Cell Sinaling Technolog, 9532, 1:1000) and conducted the preliminary experiment, but the experimental results were not satisfactory. We have also consulted with merchants that PARP antibodies are not fully applicable to sheep-specific antibodies. So while we would like to complement the experiments, we have not found any available antibodies. Secondly, in our experiment, we mainly tested the correlation indexes of BAX, BCL2, CASP3, and CASP9 with respect to apoptosis, and found that the BAX/BCL2 ratio decreased significantly after overexpression of CREB1, which is also consistent with the results of previous studies. We think that our data may not be optimal, but should be sufficient to conclude a conclusion that CREB1 inhibits apoptosis in endometrial cells.
Regarding autophagy, we performed immunofluorescence experiments with the previous LC3 antibody and observed no corresponding positive staining.The probable reason, in my opinion, is that the autophagy level of healthy cells is relatively low, and it may be necessary to carry out corresponding experiments using autophagy triggers and starvation treatments. However, the preliminary exploration of this experiment and the re-purchase of siRNAs inducers requires a certain amount of time, and our revised time is actually very short. In addition, the reason why we did not proceed with a corresponding experiments on LC3B-GFP or other stanning for endogenous LC3 redistribution at that time is that we have also detected other autophagy-related antibodies, such as P62, ATG family, etc. In our experimental results, it can be seen that CREB1 has a larger effect on P62 and ATG7. And, we mentioned in the discussion that there is a literature report that CREB1 mainly affects the expression of ATG7. From our experimental results, it can be seen that interference with CREB1 promoted autophagy.
While we are unable to conduct these specific experiments within the given time frame for revisions, we agree that measuring PARP cleavage and LC3 redistribution would provide more direct assessments of apoptosis and autophagy. In light of the reviewer's feedback, we have added a discussion of the limitations of our current approaches, and have highlighted these assays as important directions for future studies to clarify the underlying mechanisms. We sincerely thank the reviewer for these thoughtful suggestions which will significantly strengthen our ongoing and future work in this area. We hope that the added discussion acknowledging current limitations and identifying next steps will sufficiently address the reviewer's concerns at this stage, but we welcome any additional feedback about how to further improve the manuscript within the constraints of the current study.

Reviewer 2 Report
The presented Manuscript entitled “CREB1 is Involved in miR-134-5p-mediated Inhibition of Endometrial Stromal Cell Proliferation, Apoptosis and Autophagy” supports several interesting results considering the potential involvement of CREB1 protein in the process of uterine receptivity establishment. The authors of the manuscript presented an impressive amount of results obtained during multiple of analyses. The manuscript is written very well, however there is some issues that must be corrected.
Line 101 and similar: Please use abbreviation “Fig.” instead of Figure across the manuscript.
Line 119: The Authors missed the results for one comparison.
Line 126 and similar: in some cases, there is comma missing in large number, please correct across the manuscript.
Line 136 and similar: There is big problem with the numbers of all tables across the manuscript. Please insert the correct numbering of tables into the text, because without it it is very difficult to refer to the correct data in supplementary files.
Line 165 and similar: I suggest replacing “()” in the all figures description and use ":" instead of them to make the descriptions easier to read. For example, "Characteristics of CREB1 during estrous cycle of sheep. A: The qRT-PCR detected the 165 expression of some key node genes in endometrium of Hu sheep with different fecundity . B: The 166 mRNA abundance of CREB1 gene in the endometrium of follicular phase of Hu sheep with different 167 fertility. "
Line 166: Unnecessary space at the end of the sentence.
Line 173: Chane to “uterine”
Figure 4 and similar: The presented figures have different size, however I suppose that these will be corrected during the publication process.
Line 462: There is missing dot at the end of the sentence.
Line 478 and similar: Please standardize the use of italics across the manuscript.
Figure 9: I suggest to transfer the figure in to the graphical abstract of the publication.
Line 506: Were the period of the cycle confirmed by any additional methods (P4 level of ovarian morphology)?
Line 517-530: Please rewrite this fragment in more descriptive form.
Line 543: Change “u” for micro symbol.
Line 548: Please attach the sequencing specification (number of pair-ended reads and sequencing depth).
Line 551 and similar: Please attach the version of the software used for analyses.
Line 560: Please attach reaction conditions.
Line 568: Please precise the source of primers sequences. If the authors projected them by their own, then supply the information about the software used, validation experiments and reaction efficiency. Were the results confirmed by the melting curve analysis? I'm a little bit surprised that all the tested genes qPCR reactions were run in the same conditions, that is why I am asking about validation, efficiency and melting curve analysis. Please attach the representative melting curves for all the genes in supplementary files. Please confirm that the qPCR experiments were conducted in agreement with MIQUE guidelines.
Line 573: Please precise the amount.
Line 574: Please attach the transfer method and conditions for transfer and electrophoresis.
Line 575: Please precise the blocking method.
Line 576: Please attach the antibody hosts for each position of the table.
Line 598: Please precise the magnification applied for images and the used microscope and software.
Line 603: Please attach the kits sensitivity and range as well as inter- and Iintrassay coefficients of variation.
Line 619: Please attach the reactions efficiency.
Line 624: Please attach blocking step description.
Line 628: Please attach microscope model, used software and describe negative control protocol.
Line 636: How was the cells harvested? Did the Authors used trypsin or another treatment, please specify.
Line 646: Please attach microscope model and used software.
Line 652: Please attach manufacturer country and version of the software.
Line 664: Please correct the format of the cells’ amount and specify these details for the cultures described in previous sections.
Line 673: What type of post hoc test was used? Did the Authors confirmed the normal distribution of the results and homogeneity of variance?
I reccomend minoor corrections from a native speaker.
Author Response
The presented Manuscript entitled “CREB1 is Involved in miR-134-5p-mediated Inhibition of Endometrial Stromal Cell Proliferation, Apoptosis and Autophagy” supports several interesting results considering the potential involvement of CREB1 protein in the process of uterine receptivity establishment. The authors of the manuscript presented an impressive amount of results obtained during multiple of analyses. The manuscript is written very well, however there is some issues that must be corrected.
Line 101 and similar: Please use abbreviation “Fig.” instead of Figure across the manuscript.
Thank you for kind suggestions. We have made modifications across the manuscript as requested.
Line 119: The Authors missed the results for one comparison.
Thank you for kind suggestions. We have complemented the results in the corresponding paragraphs.
Line 126 and similar: in some cases, there is comma missing in large number, please correct across the manuscript.
Thank you for kind suggestions. We have reviewed the full text and made changes.
Line 136 and similar: There is big problem with the numbers of all tables across the manuscript. Please insert the correct numbering of tables into the text, because without it it is very difficult to refer to the correct data in supplementary files.
Thank you for kind suggestions. We have corrected the number of tables into the text.
Line 165 and similar: I suggest replacing “()” in the all figures description and use ":" instead of them to make the descriptions easier to read. For example, "Characteristics of CREB1 during estrous cycle of sheep. A: The qRT-PCR detected the 165 expression of some key node genes in endometrium of Hu sheep with different fecundity . B: The 166 mRNA abundance of CREB1 gene in the endometrium of follicular phase of Hu sheep with different 167 fertility. "
Thank you for kind suggestions. We have made modifications across the manuscript as required.
Line 166: Unnecessary space at the end of the sentence.
Thank you for your suggestions. We have removed the unnecessary space at the end of the sentenceu across the manuscript.
Line 173: Chane to “uterine”
Thank you for kind suggestions. We have made modifications in the corresponding sections as required.
Figure 4 and similar: The presented figures have different size, however I suppose that these will be corrected during the publication process.
Thank you for kind suggestions. We have adjusted all figures to the same width.
Line 462: There is missing dot at the end of the sentence.
Thank you for kind suggestions. We have added commas in the corresponding sections.
Line 478 and similar: Please standardize the use of italics across the manuscript.
Thank you for kind suggestions. We have made modifications across the manuscript as requested.
Figure 9: I suggest to transfer the figure in to the graphical abstract of the publication.
Thank you for kind suggestions. We have moved Figure 9 to the graphic summary of the publication as you suggested.
Line 506: Were the period of the cycle confirmed by any additional methods (P4 level of ovarian morphology)?
Thank you for kind suggestions.For the identification of the estrus cycle: first, after the simultaneous estrus of the test ewes, we arrange for men to test the ewes on a daily basis, keeping a detailed record of the time of the ewes' estrus, and slaughtering the ewes according to this record; Second, we reconfirm by ovarian morphology at slaughter.
Line 517-530: Please rewrite this fragment in more descriptive form.
Thank you for kind suggestions. We have rewritten this section at your request.
Line 543: Change “u” for micro symbol.
Thank you for kind suggestions. We have made changes.
Line 548: Please attach the sequencing specification (number of pair-ended reads and sequencing depth).
Thank you for kind suggestions. We have attached the sequencing specification (number of pair-ended reads and sequencing depth).
Line 551 and similar: Please attach the version of the software used for analyses.
Thank you for kind suggestions. We have supplemented the software version.
Line 560: Please attach reaction conditions.
Thank you for kind suggestions. We have added the corresponding reaction conditions.
Line 568: Please precise the source of primers sequences. If the authors projected them by their own, then supply the information about the software used, validation experiments and reaction efficiency. Were the results confirmed by the melting curve analysis? I'm a little bit surprised that all the tested genes qPCR reactions were run in the same conditions, that is why I am asking about validation, efficiency and melting curve analysis. Please attach the representative melting curves for all the genes in supplementary files. Please confirm that the qPCR experiments were conducted in agreement with MIQUE guidelines.
Thank you for kind suggestions. We confirmed that the qPCR experiments were performed in agreement with MIQUE guidelines. The primers involved in this experiment were all designed by Primer 5.0 software. In general, we designed at least 3 primers for each gene, and blast software was used to determine the specificity of all primers. The fragment sizes of all genes were also confirmed by agarose gel electrophoresis. We selected the most appropriate primer sequence of primers under uniform conditions. Following your suggestion, we have also attached representative melting curves for all genes for your reference.
Line 573: Please precise the amount.
Thank you for kind suggestions. We have added the precise amount.
Line 574: Please attach the transfer method and conditions for transfer and electrophoresis.
Thank you for kind suggestions. We attached the transfer method and the conditions for transfer and alectrophoresis.
Line 575: Please precise the blocking method.
Thank you for kind suggestions. We blocked the protein bands with 5% non-fat milk in TBST, and we have added this description in the manuscript.
Line 576: Please attach the antibody hosts for each position of the table.
Thank you for kind suggestions. We have added the specific antibody host information for each position of the Table S17.
Line 598: Please precise the magnification applied for images and the used microscope and software.
Thank you for kind suggestions. For this experiment, a Pannoramic MIDI digital sliDE scanner was used, which can scan the processed tissue slices, and Slideviewer software can adjust any viewing multiple of the slices. We have also added specific descriptions of the scanning system and the graphics acquisition software at the corresponding positions in the manuscript.
Line 603: Please attach the kits sensitivity and range as well as inter- and Iintrassay coefficients of variation.
Thank you for kind suggestions. We have attached the kits sensitivity as well as inter- and Iintra-ssay coefficients of variation.
Line 619: Please attach the reactions efficiency.
Thank you for kind suggestions. In general, plasmid transfection efficiency above 50% is called easy transfection. Lipofectamine 3000 is widely used in gene deletion and acquisition experiments as an economical and practical transfection reagent.Therefore, the endometrial stromal cells used in our experiment also achieved a transfection effect of over 50% when transfected with Lipofectamine 3000. Therefore, the endometrial stromal cells used in our experiment also achieved a transfection effect of over 50% when transfected with lipo3000. In addition, as shown in Fig. 2D-2E, we repeat many tests and the interference efficiency and the over-expression efficiency of the siRNA are very stable, which can be used for later experimental studies. We have also added the corresponding paragraphs of material and methods for additional explanation.
Line 624: Please attach blocking step description.
Thank you for kind suggestions. We have attached the blocking step description.
Line 628: Please attach microscope model, used software and describe negative control protocol.
Thank you for kind suggestions. We have attached microscope model, used software and describe negative control protocol.
Line 636: How was the cells harvested? Did the Authors used trypsin or another treatment, please specify.
Thank you for your suggestions. I wonder if you are asking about the method of cell collection in the EdU experiment. If so, we didn't use pancreatic enzymes or other methods to collect cells. As described in our material approach, we first plated on glass coverslips in a 24-well plate on which the inoculated cells were grown. After treating the cells for 24 h, strict experiments were carried out according to the instruction manual of the edu kit. Finally, the cell slides were taken out and placed on the slides, and the pictures were taken under the microscope to save the pictures.
Line 646: Please attach microscope model and used software.
Thank you for kind suggestions. We have attached microscope and used software.
Line 652: Please attach manufacturer country and version of the software.
Thank you for kind suggestions. We have attached manufacturer country and version of the FlowJo software.
Line 664: Please correct the format of the cells’ amount and specify these details for the cultures described in previous sections.
Thank you for kind suggestions. We have corrected the format of the cells’ amount, and we also specify these details for the cultures described in previous sections.
Line 673: What type of post hoc test was used? Did the Authors confirmed the normal distribution of the results and homogeneity of variance?
Thank you for kind suggestions. We apologize for the lack of clarity in the description of statistical analysis and we would make the following explanation: first, the normality of the data was checked using the Kolmogorov Smirnov goodness-of-fit test and it is found that most of the data conform to normal distribution; second, the student’s t test (two groups) and one-way analysis of variance (ANOVA) were used for single comparison and multiple group comparisons, respectively. The results showed that only a very small number of data did not conform to variance homogeneity. When the homocedasticity of the data did not meet the requirements, nonparametric test were used, specifically the Mann-Whitney U test for two group comparisons or the Kruskal-Wallis H test with Dunn's post-hoc test for multiple groups Additionally, we added more specific description in the part of statistical analysis, and we hope to meet your requirements after modification.
